# Application of ATR-FT-MIR for Tracing the Geographical Origin of Honey Produced in the Maltese Islands

**DOI:** 10.3390/foods9060710

**Published:** 2020-06-01

**Authors:** Jean Paul Formosa, Frederick Lia, David Mifsud, Claude Farrugia

**Affiliations:** 1Department of Chemistry, University of Malta, 2080 MSD Msida, Malta; jean.p.formosa.12@um.edu.mt (J.P.F.); claude.farrugia@um.edu.mt (C.F.); 2Department of Rural Sciences and Food Systems, University of Malta, 2080 MSD Msida, Malta; david.a.mifsud@um.edu.mt

**Keywords:** Malta, honey, PLS-DA, LDA, SVM, FF-ANN, chemometrics, ATR-FT-MIR

## Abstract

Maltese honey has been produced, marketed, and sold as an exclusive local gourmet food product for countless years. Yet, thus far, no study has evaluated the individuality of this local food product. The evaluation of the parameters and properties which characterise the provenance and floral source of honey have been the subject of various studies worldwide, owing to the price and potential beneficial properties of this food product. Models analysing the potential of attenuated total reflection mid-infrared (ATR-FT-MIR) spectroscopy in discriminating and classifying local honey from that of foreign origin were investigated using 21 Maltese honey samples and 49 honey samples collected from abroad (Sicily, Greece, Sweden, Italy, France, Estonia and other samples of mixed geographical origin). Through a combination of spectroscopic techniques, spectral transformations, variable selection and partial least squares discriminant analysis (PLS-DA), chemometric models which successfully classified the provenance of local and non-local honey were developed. The results of these models were also corroborated with other classification and pattern recognition techniques, such as linear discriminate analysis (LDA), support vector machines (SVM) and feed-forward artificial neural networks (FF-ANN).

## 1. Introduction

There is a considerable number of apiaries in Malta and Gozo, producing honey which is sought by locals and tourists alike, and valued for its unique taste and characteristics. At the moment, there is a sizable number of beekeepers selling Maltese honey directly or through local markets. However, the vast amount of honey being sold is raising suspicion that there might be cases of fraud, where the honey is either being mislabelled as Maltese honey, or else adulterated with sugar syrup and/or non-local honey.

The sugars fructose and glucose account for about 85% of honey solids, given that floral nectar is the source of honey sugars. Glucose and fructose are reported to be the only monosaccharides in honey, with an average concentration of 38% *w*/*w* for fructose and 31% *w*/*w* for glucose [1,2]. Oligosaccharides represent about 10% of the total honey weight [3] and are composed of several units, generally two to six units of glucose and fructose, with glycosidic linkages in different positions. Siddiqui [2] characterised 14 disaccharides and 11 trisaccharides, while Doner [4] showed that there is satisfactory evidence for the presence of 10–13 disaccharides and 8–9 trisaccharides. More recent studies have shown that 25 trisaccharides and 10 tetrasaccharides have been found in honey samples from Spain and New Zealand [5]. The composition of oligosaccharides in honey is related to the floral source, however, it is difficult to specify one sugar as a floral marker for honey. It has been suggested that the ratio of certain sugars, along with other parameters, can be used to differentiate between honeys [6].

There is a considerable number of apiaries in Malta and Gozo, producing honey which is sought by locals and tourists alike, and valued for its unique taste and characteristics. Malta has a long history with honey. Some claim that the name of the island is derived from the Greek word “Μελίτη” (Melite) meaning ‘honey-sweet’, and furthermore, ancient bee hives dating to the time when the Romans occupied Malta (c. 200 BC) have been found. However, in recent times, there have been several concerns about the authenticity of Maltese honey being sold at local markets. At the moment, there are a sizable number of beekeepers selling Maltese honey directly or through local markets. However, the vast amount of honey being sold is raising suspicion that there might be cases of fraud, where the honey is either being mislabelled as Maltese honey or else adulterated with sugar syrup and/or non-local honey. Maltese honey is mostly collected by the Maltese honeybee *Apis mellifera ruttneri*, which is indigenous and endemic to the Maltese islands [7,8]. However, in recent years, this endemic species has become under threat, due to the importation of foreign queen bees from Sicily. While there is no published evidence on the effect of the mixing of the two honeybees, *Apis mellifera ruttneri* and *Apis mellifera sicula*, on the chemical and physical properties of Maltese honey, there are still significant threats to the local bee community and beekeepers. Firstly, there is the risk of the loss in biodiversity due to the wiping out of the indigenous honeybee, and furthermore, there is also a risk to the general public, due to the possible aggressive behaviour of the resulting hybrids.

Published literature on Maltese honey is lacking, and up till now, no in-depth chemical profiling has been performed on Maltese honey; in fact, the literature is mainly based on physicochemical parameters. These studies include HMF content, diastase and proline levels, and total phenolic content [9,10,11]. More recently, a comprehensive study with regards to a number of physicochemical parameters and sugar composition has also been published [11]. Furthermore, little or no comparison has been performed with regards to the physicochemical and chemical properties of Maltese honey with honey samples from other regions of the world.

Infrared (IR) spectroscopy is a technique used frequently in food analysis for authentication, quantification and detection of adulteration [12], and has been favoured as a rapid, non-destructive, cheap and reagent free technique in the food industry [13,14]. In combination with chemometrics, IR spectroscopy was successfully applied for the determination of different attributes and adulterants [15] in several food samples, including juice analysis [16]; alcoholic beverages [7,17,18,19] must and wine analysis [9,20,21], polymethoxylated flavone of orange oil residues [22], organic acids and carbohydrates determination in fruits [23], and characterisation of olive oil and olive pulp [24,25].

Near-infrared (NIR) and mid-infrared (MIR) methods, coupled with signal processing and chemometric techniques, have been extensively developed in recent years, for quality control and the authentication of honey samples [26], including numerous attenuated total reflection mid-infrared (ATR-FT-MIR) methods. Several methods have been developed for the detection of sugar syrups in honey, particularly using partial least squares discriminant analysis (PLS-DA) [20,27,28,29,30,31,32,33], and through the use of principle component analysis (PCA), linear discriminate analysis (LDA) and artificial neural networks (ANN) [34]. PLS regression and principal component regression (PCR) have also been use in conjunction with ATR-FT-MIR spectroscopy for the quantification of sugars in honey; namely, glucose, fructose, sucrose, and maltose, melezitose and turanose [35,36,37,38,39].

ATR-FT-MIR methods have also been successfully employed in conjunction with chemometric methods, including PCA and LDA, for differentiation of botanical origin [36,37,38,39,40,41,42]. Hennessy et al. [40], used several signal processing methods on MIR spectra of Corsican and non-Corsican honey, in conjunction with FDA (factorial discriminant analysis) and PLS analysis for classification. NIR spectroscopy and signal processing methods were also shown to be essential tools in conjunction with SIMCA (soft independent modelling of class analogy) and PLS for the geographical classification of Irish, Mexican, and Spanish, Argentinean, Czech, Hungarian, and Irish honey samples [41,42]. These studies all employ the use of spectral transformations prior to multivariate analysis. Spectral transformations are especially important when applied to IR data, in order to remove spectral artefacts such as baseline shifts and multicollinearity and can also reveal ‘hidden’ information by emphasising small spectral variations. The aim of this research is the analysis of Maltese and foreign honey by ATR-FT-MIR, alongside several data treatment and pattern recognition techniques. The study also aims to identify which spectral transformation or combination of them are more adequate for their discrimination. 

## 2. Materials and Methods 

### 2.1. Honey Samples

A total of 70 samples were collected and 21 local samples were directly collected from Maltese and Gozitan beekeepers post honey harvest, between the period of 2015 and 2016. Further details are presented in the Appendix A. Overall, 49 foreign samples were collected from different Mediterranean countries directly from various international beekeepers associations. Furthermore, foreign samples sold from local supermarkets were also included. All samples were kept in the dark at 20 °C until analysis. 

### 2.2. FTIR Method

Prior to scanning, honey samples were homogenized after heating to 30 °C for one hour, followed by stirring. A Shimadzu IR-Affinity 1 equipped with a Silver Gate Zn/Se ATR was used for spectral acquisition. The instrument was set to acquire 32 scans per spectrum at a resolution of 4 cm^−1^ in the range of 4000−550 cm^−1^. In order to obtain a spectrum with a high signal to noise ratio and to reduce the error in the baseline, the instrument was blanked before each sample and each spectrum was run in triplicate. The mean of these replicates was then used in the following data analysis procedures. 

### 2.3. Chemometric Analysis

Initial data treatment included first removing the region between 2800 and 1800 cm^−1^ in which no peaks arise. The honey sample matrix contains a negligible amount of chemicals which have active bands in this region [29]. The spectra were also trimmed at the ends to a range of 740–3600 cm^−1^, in order to remove regions which contained a significant amount of noise and no relevant chemical data (Figure 1). The spectra obtained were subjected to different spectroscopic signal processing techniques which were evaluated and compared. These include subtraction of a linear baseline, multiplicative scatter correction (MSC), orthogonal signal correction (OSC), standard normal variate (SNV), and first and second derivative Savitzky–Golay transformations. The effect of the different spectral transformations on the final classification outcomes was compared to those obtained without any signal processing.

Several spectral transformations were applied prior to statistical analysis using the Unscrambler X (CAMO A/S, Oslo, Norway). Smoothing was the first transformation applied to the IR spectra. There are a variety of smoothing algorithms which can be applied to spectra, including moving average, Gaussian, median and Savitzky-Golay. The spectrum with maximum smoothness and minimum distortion from the original signal was selected, thus, a compromise between noise reduction and retention of information was evaluated. This was also further confirmed by PLS-DA analysis, which showed that the best improvement and highest explained variance out of all smoothed spectra was obtained when using a median filter with a gap size of three.

MSC, OSC, detrending, deresolving, SNV, along with a combination of SNV and detrending filters were then applied to the smoothed spectral data, in order to determine their effect on the misclassification rate and the RMSE error. Furthermore, first and second Savitzky-Golay derivative transformations were applied to the spectra in the region with a gap size of 7 points and a polynomial order of two. 

Principal component analysis (PCA) was carried out on the data, in order to hint at possible outliers or any possible clustering of the Maltese samples present within the data set. The supervised chemometric treatment was performed using PLS-DA, in order to classify the geographical origin with regards to Maltese and non-Maltese samples. The former samples were assigned a dummy variable of 1, while the latter were assigned a value of 0. Samples with a predicted value of >0.5 were thus labelled as foreign, while the remaining samples were labelled as local. PLS-DA analysis was carried out on the whole data set, using leave one out cross-validation (LOOCV), after which PLS-DA was repeated using excluded rows validation (ERV), with the exclusion of one third of the samples from each class to assess for model overfitting. The RMSE for the model was calculated as shown below, in order to further assess the accuracy of the model. Where *y*_pred_ corresponds the value between 0 and 1 generated by the model, whilst *y*_ref_ corresponds to the dummy variable to which the honey was assigned.
$$\mathrm{RMSE}=\sqrt{\frac{\sum_{i=1}^{n}{\left(y_{\mathrm{pred}}-y_{\mathrm{ref}}\right)}^{2}}{n}}$$

The optimum model for each transformation was chosen after an assessment of the PLS-DA model parameters. The classification accuracy of the LOOCV and ERV models, explained as X and Y variance, RMSE and number of factors were used to evaluate the performance of the chemometric models. 

#### 2.3.1. Variable Selection

Once the optimum number of factors is determined, the data points which had a VIP (variable importance in projection) >0.8 were then used to develop subsequent PLS-DA models. The VIP score is a measure of a variable’s importance in the PLS model. It represents the contribution of a variable to the PLS model and is determined through a weighted sum of the squared correlations between the model components and the original variable. A value of less than 0.8 is typically considered to be a small VIP, and thus, a candidate for deletion from the model [43]. VIP scores are useful in understanding X space predictor variables that best explain y variance. VIP scores give an estimate of the contribution of a given predictor to a PLS regression model [44].

Furthermore, stepwise linear canonical discriminant analysis (SLC-DA), as implemented within JMP, was also used, in order to reduce the number of variables used in PLS-DA models. A stepwise analysis allows for the manual selection of variables used to build the linear model up to a maximum number of entries (*n*–1), where n is the number of samples in the sample set. The model containing the most discriminant variables was selected on the basis of a low F-ratio and a high *p*-Value.

#### 2.3.2. Statistical Analysis

Feed-forward artificial neural networks (FF-ANN), support vector machines (SVM) and linear discriminant analysis (LDA) were implemented as a further corroboration and validation to the PLS-DA models. FF-ANN, LDA and SVM analysis were carried using a Python script and the ‘scikit-learn’ Machine Learning toolbox for Python [45]. FF-ANN models were implemented on data without variable selection, whilst SVM and LDA classification methods were applied on data with SLC-DA model selection. 

SVM models have no limit on the number of variables which can be used in a model. Nonetheless, SVM models require a computationally intensive grid search and thus analysis were performed on SLC-DA selected variables. Models for LDA were also performed on the SLC-DA selected variable, as this classification technique is usually limited to small number of variables, which must be less than the number of samples in each class. On the other hand, FF-ANN models are suited for modelling data with a large number of variables, and thus were used to model data with no variable selection. In all cases the models were validated both using ERV in a similar fashion to PLS-DA models.

The aforementioned statistical analysis and variable selection steps were also carried out on the fingerprint region (760–1400 cm^−1^), in order to determine the effect of using this portion of the spectrum only on the prediction rate and RMSE of the models.

## 3. Results and Discussion 

### 3.1. Geographical Classification Using ATR-FT-MIR

Monosaccharides, water, and other sugars are the main components in honey, thus, most of the spectral peaks observed in honey IR spectra appertain to vibrational modes exhibited from sugars and water [23]. Water in honey shows up as a very distinct broad peak between 3500−3000 cm^−1^ in the honey IR spectrum (Figure 1). Additionally, a peak is observed between 3000−2800 cm^−1^, which arises from vibrational modes of carbohydrates [32], carboxylic acids [46] and amino acids [29]. The region between 1700−1600 cm^−1^ shows the vibrational modes from water [47], carbohydrates [32] and the amide I band [48]. The peaks within the fingerprint region (1500−700 cm^−1^) are attributed to various vibrational modes of carbohydrates and ketones. The vibrations that occur between 1200 and 1300 cm^−1^ are attributed to the presence of –C–O bonds, whilst that at 1750 cm^−1^ accounts for the carboxylic acid functionalities (C=O) of various carbohydrates. The bands observed in the range between 1150 and 995 cm^−1^ are attributed to the stretching and bending vibrations of C–O, C–H and C–OH vibrations arising from carbohydrates. [34,36].

The first data handling stage involved the removal of the region between 2700 cm^−1^ and 1800 cm^−1^, as it contained no IR bands which are expected to show up from the honey sample matrix; for simplicity’s sake, this region will be referred to as the ‘whole spectrum’ henceforth. The median filter transformation was particularly effective in removing any noise generated by the ATR-FTIR while leaving any slight spectral variations intact. Further spectral transformations were then applied to median filter smoothed MIR spectra, since the application of some spectral transformations such as derivative transformations tended to accentuate any noise present. 

A visual inspection of the MIR spectra (Figure 1) and the resulting spectral transformations revealed no regions which offer discrimination between Maltese and non-Maltese samples. Furthermore, PCA identified no outliers within the dataset for the untreated spectra, or when the spectra were subjected to spectral transformations. Through PCA, no samples were observed to cluster according to geographical origin. This result is not unexpected, since the honey samples being tested do not differ only by geographical origin, since other sources of variability, such as botanical origin, were present. PCA analysis is presented in Appendix A. 

#### 3.1.1. PLS-DA and Variable Selection

PLS-DA was used as the primary statistical model for sample classification and the prediction of Maltese and non-Maltese samples (Table 1). In the majority of the cross validated PLS models on the ‘whole’ MIR spectra (Table 1), no samples were misclassified, except for the second derivative transformation model, which exhibited an accuracy of 98.6%. The RMSEs for all the LOOCV models were considerably low, wherein the RMSE will effectively describe the average distances of the predicted sample towards the dummy classification system used. 

ERV PLS-DA models exhibited a decrease in the % accuracy, accompanied with an increase in the RMSE; this is expected, since the models relied on smaller number of samples. Ideally, well-calibrated models should exhibit little or no change on moving from cross-validated models to excluded rows validated models, whereas a significant drop in accuracy and an increase in the RMSE often suggests that the model is over-fitted. An over-fitted model will not solely describe the systematic variation in a model, but will also describe some of the random variation within the dataset and will give inaccurate predictions. For most of the models (Table 1), the drop in prediction accuracy was not very large, since most models show an accuracy >95%, which suggests that these models were well-calibrated.

The use of VIP scores for data reduction in PLS shows a notable improvement with regards to the classification rate in both the cross validated and excluded rows models. Furthermore, a markedly larger improvement was also observed when using SLC-DA for variable selection (Table 1), where all the internally validated PLS-DA models exhibited no misclassifications and considerably lower RMSEs than the PLS models without variable selection and VIP variable selection. An example of such a plot of VIP and SLCDA scores for the media transformation is included in Figure 2b,c. 

The externally validated PLS-DA models also showed a significant improvement when SLCDA is used, with most models showing no misclassifications. The PLS-DA model improvement can be attributed to the large amount of variable reduction, from around 550 variables to around 20–40 variables, wherein the amount of redundant and collinear variables is reduced (Figure 2). 

Generally, there is a marked improvement when only the fingerprint region was used to develop PLS-DA model (Table 1), when compared to models on the ‘whole’ spectra. A similar trend towards a drop in the prediction accuracy for the excluded rows validation is also observed in this case. The OSC, SNV and a combination of SNV and detrending transformations generated PLS-DA models, which correctly classify all the samples through external validated models, without any variable selection as show in Figure 3. This highlighted the effectivity of these transformations when used in conjunction with PLS-DA for the geographical profiling of Maltese and non-Maltese samples.

The removal of the variables which had VIP scores less than 0.8 on the fingerprint region generally decreased the RMSE in the internally validated PLS-DA models, which maintained their classification accuracy. The PLS-DA models using external validation were also shown to generally improve, whereas the models using OSC and median filter transformations showed an increase in the number of misclassifications. Moreover, PLS-DA models using the variables selected by SLC-DA (Table 1) generally showed a marked improvement on the models using no variable selection and VIP variable selection, except for the model using a de-resolve transformation.

Furthermore, apart from a higher accuracy and lower RMSE, PLS-DA models on spectral transformations of the fingerprint region generally exhibited a higher % explained variance for the predictor matrices when compared to the ‘whole’ spectra. This was generally true in the case of PLS-DA models without variable selection and models which used VIP scores for variable selection. Nonetheless, the performance of the PLS models using the SLC-DA variables was similar in both cases (Table 1). The lack of improvement over using the fingerprint region versus the whole spectrum for analysis in the case of SLC-DA, is due to the fact that SLC-DA is very effective at variable selection, thus removing any redundant variables present in the region from 3600−2800 cm^−1^. 

In light of these findings, it can be concluded that the fingerprint region is more suited for the differentiation of local and foreign samples using PLS-DA analysis, since it carries more relevant information and still gives very good classification accuracy without the need of variable selection. Lastly, while the model parameters give a good indication of the performance of the PLS-DA models, at this stage, they should not be used to single out the best performing transformation method for classifying Maltese and non-Maltese honey using PLS-DA. This is because the samples only represent a small set of local honey and an even smaller set of non-local honey, and thus different samples might be better represented using different transformations. Nevertheless, the high classification rates and low error values highlight the potential application of ATR-FT-MIR spectroscopy and spectral transformations in combination with PLS-DA for the routine classification of local and foreign honey.

#### 3.1.2. Other Models 

Excluded row FF-ANN models (Table 2) on spectral transformations of the whole MIR spectra showed a slight improvement in the classification rate when compared to their respective PLS-DA models. Conversely, the excluded rows validated FF-ANN models performed on spectral transformations of the fingerprint region showed an increase in the number of misclassifications for nearly all the spectral transformations, except for models derived from MSC and SNV transformed spectra. This infers that FF-ANN models were more effective at extracting information from the ‘whole’ spectra for classification than from the fingerprint region. A similar trend was observed in the excluded rows SVM model results (Table 3), where the classification rate was generally higher for SVM models of spectral transformation on the ‘whole’ spectra than for spectral transformation in the fingerprint region; thus, the same inference can be made.

Nevertheless, the high classification rates obtained by most excluded rows validated FF-ANN, SVM and PLS-DA models, further corroborating the use of MIR spectra and spectral transformations as a method for the classification of local and foreign samples. The PLS-DA model results were also further corroborated by the LDA model results (Table 3), which showed no misclassifications in most instances. In fact, LDA models commonly showed better classification rates than the corresponding PLS-DA models. These results highlight the potential use of open source pattern recognition packages for further development and implementation in chemometric applications.

## 4. Conclusions

Most spectral transformations on ATR-FT-MIR data in combination with PLS-DA were shown to be very effective in classifying local and non-local honey samples. Furthermore, the use of the fingerprint region for classifying samples was shown to be more effective in PLS-DA models using no variable selection and VIP variable selection. The use of SLC-DA for variable selection was also shown to be significantly effective in decreasing the number of misclassifications, both when using the ‘whole’ spectrum and when using the fingerprint region. 

FF-ANN, SVM and LDA models were shown to offer similar classification rates to PLS-DA models and this thus corroborates the results obtained from the PLS-DA models and places confidence in the use of ATR-FT-MIR methods in conjunction with spectral transformations, for the classification of Maltese and foreign honey samples. These results highlight the potential of these methods to be further developed, for the detection of adulteration and for more in-depth profiling and classification of Maltese honey. Furthermore, the results obtained highlight the effectiveness of chemometric and pattern recognition-based approaches, in order to quickly and reliably test the authenticity of honey samples. These promising results should thus serve as an incentive for more research to be done on developing a more extensive model, using other techniques such as fluorescence spectroscopy and NMR.

## Figures and Tables

**Figure 1 foods-09-00710-f001:**
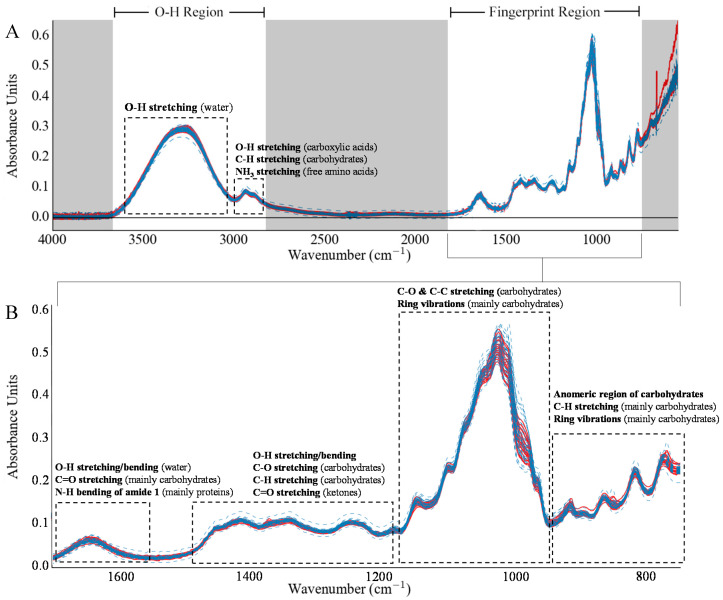
(**A**) IR Spectra of all tested samples (continuous red lines represent local samples and dashed blue lines represent foreign samples); regions which were not used in this study are shaded in grey. (**B**) Expansion of the fingerprint region highlighting the major peaks identified in this study.

**Figure 2 foods-09-00710-f002:**
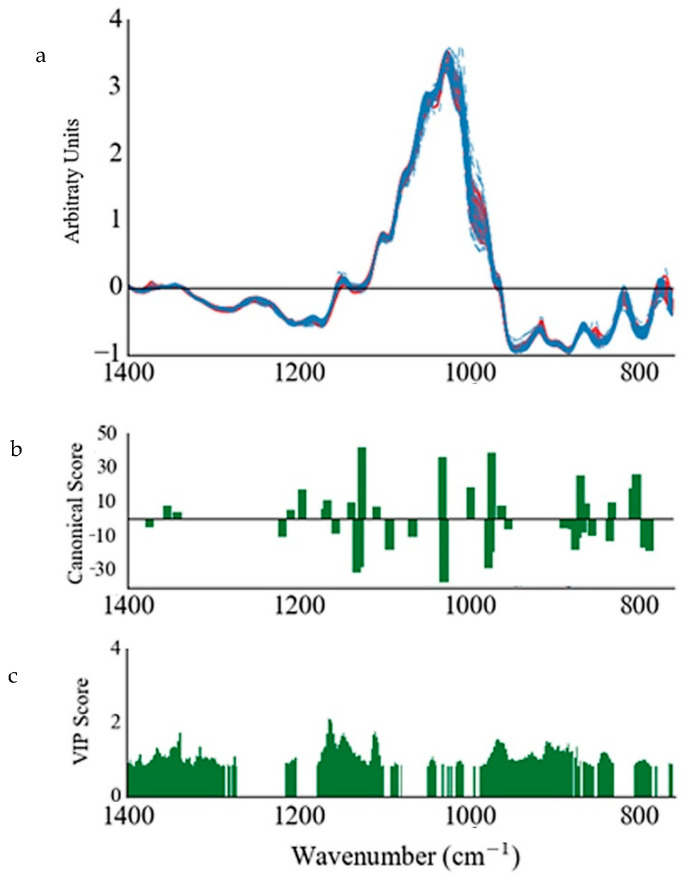
(**a**) SNVDT transformed ATR-FT-MIR spectra of fingerprint region of all honey samples (continuous red lines represent local samples and dashed blue lines represent foreign samples), (**b**) stepwise linear canonical discriminant analysis (SLC-DA) canonical scores and (**c**) VIP scores (>0.8) obtained from the variable selection procedures performed on the transformed spectra in (**a**).

**Figure 3 foods-09-00710-f003:**
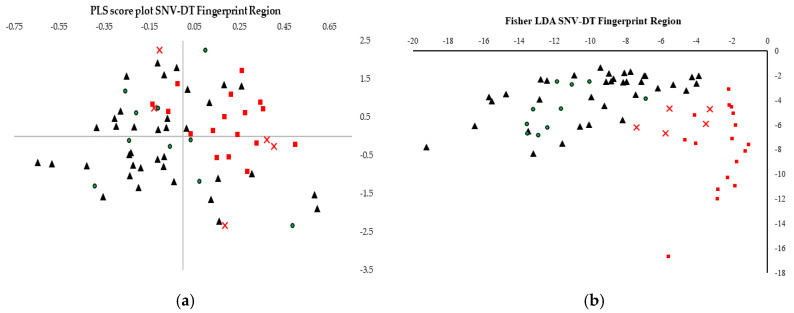
(**a**) the PLS score plot obtained using SNVDT transformed ATR-FT-MIR complete spectra under ERV. (**b**) Linear discriminate analysis performed on the extracted PLS scores. (▲) Foreign honey samples used in the training set (⬛); Maltese honey samples used in the training set (●); foreign honey samples used in the validation set (x); Maltese honey samples used in the validation set. Maltese samples are depicted in red, whilst the foreign samples are depicted in black.

**Table 1 foods-09-00710-t001:** Results from partial least squares discriminant analysis (PLS-DA) models applied to spectral transformations of attenuated total reflection mid-infrared (ATR-FT-MIR) spectra and variable selection procedures. (MF = Median Filter, 1st = Savitzky–Golay first derivative, 2nd = Savitzky–Golay second derivative, DR = De-resolve, DT = De-trend, MSC = Multiplicative Scatter Correction, OSC = Orthogonal Signal Correction, QN = Quantile Normalise, SNV = Standard Normal Variate, SNVDT = combination of SNV and DT transformations **#**F = number of latent factors extracted from the PLS model).

		Whole Spectrum					Fingerprint Region				
Variable Selection	Pre-Treatment	#F	LOOCV		ERV		#F	LOOCV		ERV	
			Accuracy %	RMSE	Accuracy %	RMSE		Accuracy %	RMSE	Accuracy %	RMSE
None	MF	14	100.0	0.096	95.7	0.201	10	100.0	0.157	98.6	0.177
	1st DSG	5	100.0	0.120	92.8	0.248	8	100.0	0.100	97.1	0.192
	2nd DSG	4	98.6	0.166	95.7	0.261	6	100.0	0.184	97.1	0.257
	DR	14	100.0	0.170	97.1	0.205	10	97.1	0.214	100.0	0.218
	DT	9	100.0	0.199	92.8	0.258	10	100.0	0.123	98.6	0.123
	MSC	12	100.0	0.101	95.7	0.211	11	100.0	0.101	98.6	0.170
	OSC	13	100.0	0.110	95.7	0.197	9	100.0	0.186	100.0	0.193
	QN	8	100.0	0.135	94.2	0.222	9	100.0	0.111	97.1	0.186
	SNV	10	100.0	0.101	98.6	0.210	10	100.0	0.126	100.0	0.169
	SNVDT	9	100.0	0.160	97.2	0.219	10	100.0	0.114	100.0	0.167
VIP	MF	15	100.0	0.047	100.0	0.111	12	100.0	0.106	97.1	0.171
	1st DSG	6	100.0	0.089	97.1	0.168	10	100.0	0.069	97.1	0.181
	2nd DSG	4	98.6	0.166	95.7	0.215	8	100.0	0.148	97.1	0.224
	DR	14	100.0	0.167	98.6	0.193	11	97.1	0.204	100.0	0.220
	DT	14	100.0	0.034	94.2	0.259	13	100.0	0.072	100.0	0.163
	MSC	14	100.0	0.095	97.1	0.181	14	100.0	0.060	100.0	0.170
	OSC	13	100.0	0.104	95.7	0.178	12	100.0	0.111	95.7	0.174
	QN	13	100.0	0.025	95.7	0.187	10	100.0	0.091	97.1	0.180
	SNV	10	100.0	0.176	98.6	0.183	10	100.0	0.125	100.0	0.164
	SNVDT	12	100.0	0.073	94.2	0.227	10	100.0	0.111	100.0	0.163
SLCDA	MF	15	100.0	0.089	100.0	0.133	15	100.0	0.094	100.0	0.149
	1st DSG	15	100.0	0.047	100.0	0.152	15	100.0	0.077	97.1	0.201
	2nd DSG	14	100.0	0.048	97.1	0.163	15	100.0	0.100	98.6	0.155
	DR	15	100.0	0.136	94.2	0.301	15	100.0	0.138	100.0	0.188
	DT	15	100.0	0.041	98.6	0.122	15	100.0	0.064	100.0	0.148
	MSC	15	100.0	0.089	100.0	0.128	15	100.0	0.076	100.0	0.139
	OSC	15	100.0	0.102	100.0	0.144	15	100.0	0.109	98.6	0.141
	QN	15	100.0	0.042	100.0	0.087	15	100.0	0.077	100.0	0.085
	SNV	15	100.0	0.047	100.0	0.111	10	100.0	0.126	100.0	0.169
	SNVDT	15	100.0	0.048	100.0	0.104	15	100.0	0.062	100.0	0.145

**Table 2 foods-09-00710-t002:** Summary of FF-ANN Model performance with no variable selection on the ‘whole’ spectrum and on fingerprint region only.

Data Pre-Treatment Method	Whole		Fingerprint	
	Accuracy (%)	RMSE	Accuracy (%)	RMSE
Median Filter	97.1	0.1299	97.1	0.1761
First Derivative (SG)	92.8	0.2429	95.7	0.1717
Second Derivative (SG)	95.7	0.2186	95.7	0.1915
Deresolve	100.0	0.0609	98.6	0.0791
Detrending	95.6	0.1648	97.1	0.1713
MSC	98.6	0.1019	98.6	0.1198
OSC	97.1	0.1604	97.1	0.1636
Quantile Normalise	95.7	0.1940	95.7	0.1958
SNV	98.6	0.1142	98.6	0.1175
SNVDT	98.6	0.0841	97.1	0.1427

**Table 3 foods-09-00710-t003:** Summary of SVM and LDA Model performance with no variable selection on the ‘whole’ spectrum and on fingerprint region only.

Data Pre-Treatment Method	Whole			Fingerprint		
	LDA		SVM	LDA		SVM
	Accuracy (%)	RMSE	Accuracy (%)	Accuracy (%)	RMSE	Accuracy (%)
Median Filter	100.0	0.0003	100.0	98.6	0.0842	98.6
First Derivative (SG)	100.0	0.0000	100.0	100.0	0.0128	97.1
Second Derivative (SG)	100.0	0.0000	100.0	98.6	0.1095	92.8
Deresolve	98.6	0.1205	95.7	100.0	0.0658	98.6
Detrending	100.0	0.0000	100.0	100.0	0.0000	98.6
MSC	100.0	0.0003	100.0	100.0	0.0000	98.6
OSC	100.0	0.0125	100.0	100.0	0.0003	100.0
Quantile Normalise	100.0	0.0000	100.0	100.0	0.0000	98.6
SNV	100.0	0.0000	100.0	100.0	0.0000	100.0
SNVDT	100.0	0.0000	100.0	100.0	0.0000	98.6

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
