# Peer review of "Application of ATR-FT-MIR for Tracing the Geographical Origin of Honey Produced in the Maltese Islands"

_foods, 2020, doi:10.3390/foods9060710_

Round 1

Reviewer 1 Report

General comments:

The paper is interesting and nice written. The manuscript is about the application of FT-IR analysis for tracing the geographical origin of honey.

Specific comments:

Line 61: could you please specify the honey variety

Line 92: PCA – some other abbreviations were used without any explanations like in line 87: MSC, OSC – same line 94 you mentioned PLS-DA already in line 84 – please correct this.

Line 103: please explain the abbreviations y pred and y ref

Line 96: is there a specific symbol for > 0.5. same for line 97 and 110, 113.

Chapter 2.3.2. maybe you use as headline: statistical analysis instead of the abbreviations

Figure 1: Please zoom in, so the reader can see the differences in the bands for this region. Please explain other bands. You probably see a big difference between 1100 and 1000 cm-1. Please explain the differences you see there.

Table 1. what do you mean by 1st and 2nd? What does #F means?

Figure 3: what is the difference between the color black and red?

Author Response

Comments and Suggestions for Authors

General comments:

The paper is interesting and nice written. The manuscript is about the application of FT-IR analysis for tracing the geographical origin of honey.

Specific comments:

Line 61: could you please specify the honey variety

Fixed Table presented in the supplementary Material.

Line 92: PCA – some other abbreviations were used without any explanations like in line 87: MSC, OSC – same line 94 you mentioned PLS-DA already in line 84 – please correct this.

Fixed

Line 103: please explain the abbreviations y pred and y ref

Fixed

Line 96: is there a specific symbol for > 0.5. same for line 97 and 110, 113.

The >0.5 was used as a cutoff point. To the authors best knowledge no symbol exists for it.

Chapter 2.3.2. maybe you use as headline: statistical analysis instead of the abbreviations

Fixed

Please explain other bands. You probably see a big difference between 1100 and 1000 cm-1. Please explain the differences you see there Figure 1: Please zoom in, so the reader can see the differences in the bands for this region.

Fixed

Table 1. what do you mean by 1st and 2nd? What does #F means?

Fixed

Figure 3: what is the difference between the color black and red?

Fixed

Reviewer 2 Report

I consider your research of high interest to the science community, but I have some comments about the problem approach to easer the reading comprehension. For this reason, I suggest a minor revision:

Introduction:
-Please describe deeply the honey constituents (sugars, carbohydrates), during the text you talk about them but maybe readers are not aware of them
-Authors talk about Malta and Gozo honey and about its unique taste and characteristic. Please describe deeply about the differences of this honey with the rest and support it with bibliography if it is possible
-Please include some reason because it is important the development of methods for the detection of honey of different origins. How a wrong labeled could cause an economic fraud or even an allergic reaction in the consumer
-Sentence 34: please include some examples of food adulteration where IR has been employed with success
-It is necessary that you describe the acronym that you use the first time that they appear: NIR, MIR, ATR-FT-MIR, PLS-DA, PCA, LDA, ANN, please review the introduction to ensure this. In addition, some times you talk about "ATR-FT-MIR" and others of "FT-MIR-ATR", please homogenize the term
-The paragraph from line 42-59 is complicated of following, authors jump from a technique to others and sometimes come back. Please, rewrite it according to the technique used not to the chemometric technique applied, it is going to be easier to follow.
-For last, please rewrite the sentence 58-59. The aim has to be more clear, I suggest some idea but writing it as you like, just with a sense of objective. For example, "the aim of this research is the analysis of Maltese and foreign honey by FT-MIR-ATR and the study of which spectral transformation or combination of them are more adequate for their discrimination"

Materials and Methods:
-Linea 63-64. "49 foreign samples were collected from different Mediterranean countries, furthermore foreign samples sold from local supermarkets were also included". Are all of them obtained from supermarkets or how do you obtain the rest? How do you ensure their origin, by labels o by beekeepers?. Please include this information.
-Line 87. What is the meaning of MSC, OSC, and SNV?

Results:

-Line 192. It is Figure 2

Author Response

Comments and Suggestions for Authors

I consider your research of high interest to the science community, but I have some comments about the problem approach to easer the reading comprehension. For this reason, I suggest a minor revision:

Introduction:
-Please describe deeply the honey constituents (sugars, carbohydrates), during the text you talk about them but maybe readers are not aware of them

Fixed

-Authors talk about Malta and Gozo honey and about its unique taste and characteristic. Please describe deeply about the differences of this honey with the rest and support it with bibliography if it is possible

Fixed

-Please include some reason because it is important the development of methods for the detection of honey of different origins. How a wrong labeled could cause an economic fraud or even an allergic reaction in the consumer

Fixed

-Sentence 34: please include some examples of food adulteration where IR has been employed with success

Fixed

-It is necessary that you describe the acronym that you use the first time that they appear: NIR, MIR, ATR-FT-MIR, PLS-DA, PCA, LDA, ANN, please review the introduction to ensure this. In addition, some times you talk about "ATR-FT-MIR" and others of "FT-MIR-ATR", please homogenize the term

Fixed

-The paragraph from line 42-59 is complicated of following, authors jump from a technique to others and sometimes come back. Please, rewrite it according to the technique used not to the chemometric technique applied, it is going to be easier to follow.

Fixed

-For last, please rewrite the sentence 58-59. The aim has to be more clear, I suggest some idea but writing it as you like, just with a sense of objective. For example, "the aim of this research is the analysis of Maltese and foreign honey by FT-MIR-ATR and the study of which spectral transformation or combination of them are more adequate for their discrimination"

Fixed

Materials and Methods:
-Linea 63-64. "49 foreign samples were collected from different Mediterranean countries, furthermore foreign samples sold from local supermarkets were also included". Are all of them obtained from supermarkets or how do you obtain the rest? How do you ensure their origin, by labels o by beekeepers?. Please include this information.

Fixed

-Line 87. What is the meaning of MSC, OSC, and SNV?

Fixed

Results:

-Line 192. It is Figure 2

Fixed

Reviewer 3 Report

The authors provide results on discrimination of Malta honey from other samples applying ATR spectroscopy. The subject of product classification is intensively explored in the literature, so the novelty of the topic is debatable.
I have no serious objections to spectral data treatment and quality of developed models, a complete set of parameters for them are provided. The weakest point of the article is that it does not answer why Maltese honeys are separated from other samples. Comments on chemical differences between the studied samples, resulting from changes of relative components content due to their botanical and geographical origin, are expected. It is good to know which spectral features are responsible for samples’ separation.

Minor notices:
1) Fig. 1 Dominant signal in 3000-3500 cm-1 region in honey spectrum originates from water. Anyway, both fructose and glucose have obvious contributions in the stretching O-H region, so showing “Water region” in 2800(!!!)-3500 is an abuse.
2) Table 1. The size of headers should fit to the column width
3) Why use only 32 scans (measurement time ~40 sec.) and then perform spectra smoothing? Each numerical procedure can reject some subtle signals, which can be important during classification.

Author Response

Comments and Suggestions for Authors

The authors provide results on discrimination of Malta honey from other samples applying ATR spectroscopy. The subject of product classification is intensively explored in the literature, so the novelty of the topic is debatable. 
I have no serious objections to spectral data treatment and quality of developed models, a complete set of parameters for them are provided. The weakest point of the article is that it does not answer why Maltese honeys are separated from other samples. Comments on chemical differences between the studied samples, resulting from changes of relative components content due to their botanical and geographical origin, are expected. It is good to know which spectral features are responsible for samples’ separation.

Minor notices:
1) Fig. 1 Dominant signal in 3000-3500 cm-1 region in honey spectrum originates from water. Anyway, both fructose and glucose have obvious contributions in the stretching O-H region, so showing “Water region” in 2800(!!!)-3500 is an abuse.

Fixed

2) Table 1. The size of headers should fit to the column width 

Fixed

3) Why use only 32 scans (measurement time ~40 sec.) and then perform spectra smoothing? Each numerical procedure can reject some subtle signals, which can be important during classification.

Thanks for the comment aim was develop fast methods for the discrimination of geographical origin, furthermore the use of smoothing as a procedure to remove subtle signal important for classification was explored and it was shown that in fact the classification system improved rather than decreased on using smoothing due to the removal of noise.

Reviewer 4 Report

This paper from Formosa et al. describes the application of FT-MIR combined with chemometrics for the authentication of Maltese honey. Although honey has been studied already with similar approaches, this study is focusing on honey from Maltese islands. Another positive characteristic of this paper is the use of statistical methods which are not used that often in food science, such as SVM.

My comments are mainly related to the limited information provided by the authors on some topics:

  • The PCA plot should be provided, at least as supporting information
  • For variable selection, why VIP was the selected method? What about other methods (regression coefficients, weights) that are used in PLS and considered very reliable?
  • What RMSE values in Tables 1, 2 and 3 mean? Is there any reference to compare them and see if they are large or small? Is there a cut off value for RMSE to be considered acceptable or not? For example, for VIP, a cut off of 0.8 was used.
  • Line 110. Do authors mean VIP>0.8?
  • Why RMSE was selected as the metrics for validation? Is it better compared to other indicators, such as Q2?
  • More information about the interpretation of SVM is needed. For PLS it is clearer since a scores plot is provided. However, for SVM, what is the outcome of the analysis and how it is interpreted?

Author Response

This paper from Formosa et al. describes the application of FT-MIR combined with chemometrics for the authentication of Maltese honey. Although honey has been studied already with similar approaches, this study is focusing on honey from Maltese islands. Another positive characteristic of this paper is the use of statistical methods which are not used that often in food science, such as SVM.

My comments are mainly related to the limited information provided by the authors on some topics:

  • The PCA plot should be provided, at least as supporting information
  • Fixed these have been added in supplementary material
  • For variable selection, why VIP was the selected method? What about other methods (regression coefficients, weights) that are used in PLS and considered very reliable?
  • We employed the use of VIP since VIP scores are useful in understanding X space predictor variables that best explain y variance. VIP scores give an estimate of the contribution of a given predictor to a PLS regression model. We employed the use of VIP since it selected the most relevant variables for the interpretation furthermore 0.8 cut off point enabled feature selection.
  • What RMSE values in Tables 1, 2 and 3 mean? Is there any reference to compare them and see if they are large or small? Is there a cut off value for RMSE to be considered acceptable or not? For example, for VIP, a cut off of 0.8 was used.
  • The RMSE where calculated using the specified equation. The RMSE in tables shows how near is the overall predictably to the true value i.e the dummy variables coding for the geographical origin. Whilst discriminate analysis uses categorical variables and only classifies the predicted origin in 0 and 1 the use of RMSE enables us to determine which transformation offered the nearest predicted value to 0 and 1 providing insights about the model consistency. To answer your question about the RMSE cut off point it is important to recall that RMSE has the same unit as the dependent variable. It means that there is no absolute good or bad threshold, however you can define it based on your dependent variable. For a dependent variable which ranges from 0 to 1000, an RMSE of 0.7 is small, but if the range goes from 0 to 1, it is not that small anymore.
  • Line 110. Do authors mean VIP>0.8?
  • It is the cut off point which enabled feature selection as latter explained in following paragraph.
  • Why RMSE was selected as the metrics for validation? Is it better compared to other indicators, such as Q2?
  • Whilst Q2 can be used for data which is expected to show a normality distribution of the predicted value in this case the predicted value should be expected to be 0 or 1 so RMSE was deemed to be more appropriate as it collectively measures all the predicted values and how much far apart are from the real value (yref 0 or 1).
  • More information about the interpretation of SVM is needed. For PLS it is clearer since a scores plot is provided. However, for SVM, what is the outcome of the analysis and how it is interpreted?

Whilst PLS PCA LDA SIMCA PCR and other methods provide loadings and score plots many other important information which enables the researcher to define which variables and a direct way to see what is actually happening to the inputted data. In machine learning such as SMV and ANN, some algorithms are referred to as black box processes because the mechanism that transforms the input into the output is obfuscated by an imaginary box, without interference from the researcher. Such data requested from SVM and ANN is not available or else it would be far more difficult to interpret than PLS and PCA.    

Round 2

Reviewer 1 Report

Thank you for the massive correction. It is now much better. Still in Figure 1, could you please number this in A) and B). And explain the region about 1100 cm-1 where do you have these ups and downs.

Author Response

Fixed
